# Lizards, Lineage and Latitude: Behavioural Responses to Microclimate Vary Latitudinally and Show Limited Acclimatisation to a Common Environment After Two Years

**DOI:** 10.3390/biology14060622

**Published:** 2025-05-28

**Authors:** Deanne M. Trewartha, Stephanie S. Godfrey, Michael G. Gardner

**Affiliations:** 1College of Science and Engineering, Flinders University, GPO Box 2100, Adelaide, SA 5001, Australia; michael.gardner@flinders.edu.au; 2Department of Zoology, University of Otago, GPO Box 56, Dunedin 9054, New Zealand; stephanie.godfrey@otago.ac.nz

**Keywords:** hydroregulation, translocation, Tiliquini, global warming, assisted migration, Egerniinae

## Abstract

Ectotherms, including reptiles, are particularly susceptible to extinction from climate change. Reptile responses to microclimate can vary from population to population, so we need to understand how climate change affects species at the population level. Reptiles rely on attaining certain body temperatures for basic bodily function; however, increasing body temperature increases dehydration risk. Therefore, when investigating behavioural response to climate, we must look at both temperature and humidity/water availability. Here we investigated behavioural response to microclimate in three geographically varied populations of pygmy bluetongues (*Tiliqua adelaidensis*), an endangered burrow-dwelling skink endemic to South Australia. Behaviour was monitored in the wild and in a common southerly translocation site. Behaviour varied with latitude; northern lizards prioritised surface activity and were only observed to be surface-active in moderate conditions and when the burrow was humid in both the wild site and the translocation. The similarity of behaviour in both sites suggest that these lizards do not readily adjust to a new environment. Our results further suggest acclimatisation to new sites may take longer than two years for all three populations and may vary with latitude of origin. Despite this acclimatisation delay, our results indicate that these lizards may cope with translocation as a mitigation strategy in the longer term.

## 1. Introduction

Many species around the globe show measurable shifts in range, abundance and phenology due to anthropogenically induced climate change [1,2,3]. The need to understand how organisms physiologically respond to environmental change is increasing in urgency as we attempt to manage climate change impacts. These impacts will further affect the localised microhabitat conditions that organisms experience [4,5].

Ectotherms have limited metabolic and evaporative capabilities and are therefore more vulnerable to climatic changes than endotherms [6]. Reptiles are an extensive group of ectotherms, adapted to almost all habitat types; however, their typically isolated populations, narrow niches and low vagility increase their vulnerability to climate change and local extirpation through limited response mechanisms [7,8,9,10,11]. As such, an estimated one in five reptile species currently face extinction threats [9,12,13]. Numerous species show population variance in behaviour across latitude and altitude, with corresponding plasticity limits [14,15,16,17,18]. Therefore, investigating the behavioural ecology of reptile populations in the context of their localised microclimate is necessary for understanding the potential scope of their response to climate change.

Environmental conditions are integral to physiological function in reptiles [19,20]. Essential bodily functions such as digestion, reproduction and locomotion depend upon the attainment of suitable body temperatures [21]. Behavioural adjustments utilising thermally heterogeneous habitat—thermoregulation—allow substantial differentials to be achieved between ambient and body temperature to extend daily activity [19,22,23,24,25]. Temperature therefore modifies behavioural type, performance and occurrence [19,26]. When predicting species responses to climate change, we need to synthesise behaviour and environment—particularly localised microclimate—as it tempers the impacts of macroclimate [2,19,27]. Thermoregulation is well studied; however, it neglects the crucial role of hydration in reptilian survival [6,28,29,30]. Behavioural responses to humidity are found in lizards at both the species and population level [31,32,33,34,35,36]. As thermoregulation trades off with risk of evaporative loss, thermal profiles require some measure of hydro-regulation—the behavioural regulation of evaporative loss—to be truly informative [30,36,37].

Translocation—the intentional human-assisted movement of living organisms within or beyond current or historic distributions [38]—is the only viable climate change mitigation strategy for many reptilian species [25,39,40,41,42]. Translocation may be used to pre-emptively establish populations in currently cooler environments where they may survive predicted warming [8,11,43]. Moving individuals to future-suitable environments likely contributes to the generally high failure rate of translocation due to a lack of understanding of species- and population-unique behavioural changes from site-of-origin to translocated site [15,44,45,46,47]. Thus, a thorough understanding of behavioural plasticity and adaptation, particularly the links between behaviour, temperature and humidity at the population level, is vital for mitigation planning.

The South Australian endemic pygmy bluetongue lizard (*Tiliqua adelaidensis*) is an endangered, cryptic, low-dispersal habitat specialist. Limited to vacated mygalomorph and lycosid spider burrows in isolated remnant grassland pockets in South Australia’s Mid North, the pygmy bluetongue has been targeted for translocation to mitigate predicted range contraction and habitat fragmentation due to climate change and agricultural practises [11,48]. Previous investigations have shown that behavioural responses to microclimate temperature and humidity varied between two geographically distinct lineages: a lineage in the south (lower latitude) and a lineage further northwards in the middle of the current range (higher latitude) [18]. Southern lizards were active at lower temperatures and higher humidity than mid-range lizards and showed significantly less daily activity [18]. Ectotherms typically have minimal approach distance—the distance between the organism and a potential predator before flight or refuge-seeking—when temperatures are within an optimal range for performance (rapid escape) [49,50]. Approach distance assays are linked to site-specific microclimate capture conditions where lizards will prioritise surface activity in the face of a potential predator, and therefore provide insight into trade-offs between daily activity and refuge-seeking [18]. Southern lineage pygmy bluetongues allowed a human observer to approach closer as base-of-burrow humidity increased, while mid-range lineage lizards retreated into burrows at greater distances from the observer [18]. Lineage variations in approach distance were shown to persist in a common environment, indicating plasticity limits that may affect the species’ ability to adjust to predicted warming and to mitigation via translocation [18].

Building ecologically relevant behavioural profiles requires the study of free-ranging animals [51,52,53], particularly difficult in specialist, cryptic or burrow-dwelling species [20]. The pygmy bluetongue’s low vagility and small home-range, centred around a single burrow, mean that wild behaviours can be studied in situ [54,55,56,57]. Direct comparisons can be made among latitudinally distinct lineages, and links between behaviour and site-specific microclimate data determined. Understanding these links will highlight microclimate conditions that predict behaviour and whether lineage groups in this and other ectotherm species are likely to differ in their ability to adjust to warming temperatures and to conditions posed by mitigation translocation. Expanding on our previous research, this current study includes a lineage at the northern edge of the species range—at highest risk of extirpation from predicted temperature increases—and thus the most in need of urgent mitigation [11,48].

Our previous findings [18] and the distribution of these low-vagility lizards in isolated habitat pockets over a diverse latitudinal and climatic range lead us to predict that the three latitudinally distinct pygmy bluetongue lineages, one northern, one mid-range and one southern, will vary in their behavioural response to microclimate temperature and humidity. Our previous findings were based on the first season in the translocation, while the current study took place at the start of the third season when all lizards had undergone an extensive acclimation period. We therefore investigate evidence for acclimatisation to the translocation site after two activity seasons. We predict that the lizards may show prolonged (greater than a single season) acclimatisation duration or limited plasticity at the translocation site. This study aims to provide insight into the ability of the target northern lineages to adjust behaviourally to a southern translocation site to further investigate acclimatisation durations in translocated lineages, as well as to directly inform management decisions for this endangered, endemic reptile.

## 2. Materials and Methods

Three geographically isolated wild lineages were used in this study, representing the latitudinal spread of the known current pygmy bluetongue range across the Mid North region of South Australia: northern (Jamestown), mid-range (Burra) and southern (Kapunda) (Figure 1). Individuals from these three populations were also monitored in a translocation site in the south of the range near Tarlee (Figure 1). The southern site near Kapunda acts as a latitudinal control for the trial translocation. All sites had natural pygmy bluetongue populations except for the translocation site, which had no known current or historic population other than those established over 2020 and 2021 (see below). The sites consisted of open grassland that was used for sheep grazing during this study.

### 2.1. Translocation

Thirty-two lizards per source lineage (northern, mid and southern) were caught using the fishing method outlined in [54] and translocated to enclosures at the southern translocation site near Tarlee in October 2020 (from Kapunda and Burra), April 2021 (from Kapunda, Burra and Jamestown) and October 2021 (Jamestown) (Figure 1). The lizards were released into two 25 × 25 m enclosures per lineage (source lineage) (Appendix A) The naturally small home-range of the pygmy bluetongue allows for natural wild behaviour to be carried out inside these enclosures [54,55,56,57]. The lizards were at the translocation site for two active seasons before behavioural observations began (Appendix A).

### 2.2. Fieldwork

Behaviour was recorded monthly from October to December 2022. Each month, pygmy bluetongue-occupied burrows were identified using a portable endoscope (Yateks M615FM), and eleven occupied burrows were marked per site. As pygmy bluetongues select burrows that are similar in diameter to the width of their head, we selected occupied burrows of a diameter > 16 mm to restrict our results to adult lizard behaviour [54].

Approach distance was measured on three consecutive days per month. All lineage groups were measured on each of the three days, and the order of the sites was randomised. Once lizard presence was determined using binoculars (any part of the lizard visible outside the burrow), the observer approached at a pace of approximately 0.5 m/s from approximately 8 m south [18]. The distance in metres from the observer to the burrow at the moment the lizard retreated was measured [49,58], and the date, time, distance, behaviour (Appendix A) and burrow identification number were recorded.

Although it is likely that the lizards in a burrow were the same individual over time, it is also possible that different lizards occupied the same burrows asynchronously. While burrow occupants were confirmed as pygmy bluetongues, lizards were not caught for individual identification to avoid unnecessary disturbance prior to behavioural assays. Some lizards vacated their burrows during the assays and new occupied burrows were located.

### 2.3. Environmental Data

Data loggers [59] were deployed at the four sites, northern (Jamestown), mid-range (Burra) and southern (Kapunda), taking 30 min readings of temperature (°C) and relative humidity (%) during each data collection episode. Data loggers were suspended in artificial burrows consisting of a 400 mm × 25 mm diameter PVC pipe, with 38 × 2 mm diameter holes drilled down the length allowing heat and humidity transfer. Three burrows were installed per site, each containing four loggers positioned at the base, middle, top of the burrow and 100 mm above the substrate (ambient) (Appendix A) [18]. These temperature and humidity data were averaged per site and per logger position and then matched to each behavioural incident.

### 2.4. Statistical Analysis

The data were compiled in Microsoft Excel and analysed in R version 4.3.1, using RStudio 2024.12.0 [60]. As environmental variables are highly colinear, we used Principal Component Analysis (PCA) to reduce the eight environmental variables down to fewer uncorrelated variables [18]. Each environmental variable was scaled and centred (divided by two times its standard deviation) so the variables were on the same scale, and a PCA was carried out on the covariance matrix.

Generalised additive mixed models (GAMMs) were used to model wild and translocated lizard log-transformed approach distances (both modelled as Gaussian) in response to microclimate variation. PC1, PC2, and PC3 were included as smoothed continuous predictor variables (to enable the estimation of non-linear relationships), and lineage (southern/mid/northern) was included as a categorical predictor variable. We were interested in whether lizards from latitudinally diverse lineages responded differently to the environmental variables. Therefore, the effects of the PC variables were evaluated separately for each lineage. Burrow identity was included as a random effect to account for repeated measurements of the lizards at the same burrow over the study duration.

## 3. Results

### 3.1. Environmental Variables

During the study in October, November and December 2022, rainfall was well above average (Appendix A) at all four sites. Rainfall for spring 2022 (October to December) was highest in the mid-range site (297.6 mm) and lowest in the northern site (231.2 mm) (Appendix A). Ambient and burrow conditions were similar across the study sites, with the exception of base-of-burrow relative humidity which varied from a seasonal mean of 59.04% (SD = 4.40) in the southern site to 94.97% (SD = 11.14) in the northern site (Appendix A). The northern site had the most variable ambient temperature (SD = 14.0) and ambient humidity (SD = 35.97), and the translocation site had the most variable base-of-burrow humidity (Appendix A).

### 3.2. Principal Component Analysis

The principal component analysis yielded three PC axes that together represented 89.6% of the variation captured in the microclimate measurements. PC1 captured most of the variation in the microclimate measurements (63%), representing the relationship between temperature and humidity (as temperature increases, humidity decreases). PC2 (13.75%) represented the relationship between temperature and humidity inside the burrow versus ambient (as surface temperature cools and humidity rises, the burrow temperature increases and humidity decreases). PC3 (13.21%) predominantly represented humidity conditions at the base of the burrow (Table 1, Appendix A). These axes were used in generalised additive mixed models to evaluate the response of lizards to environmental variation at the source and destination sites, as well as between lineages.

### 3.3. Overview

A total of 3133 approach distance observations were taken during the three-month (October–December 2022) season: 1344 in the wild and 1789 in the translocation (Appendix A). A total of 499 unique burrows were approached, representing 499 unique lizards, 176 in the wild sites and 323 in the translocation (Appendix A). After the removal of burrows, where the lizard was not visibly surface-active, 705 approach measurements were used for analysis, 317 taken in the wild and 388 in the translocation. Approach distance ranged from 0.19 m to 8.0 m, with a mean of 2.0 m. Northern-lineage lizards represented 39% of approaches, mid-lineage lizards 37% and southern-lineage lizards 24% (Appendix A).

### 3.4. Wild Lizards

During the approach distance assays on wild lizards, the northern lizards were observed on the surface at the broadest range of surface temperatures (15–45 °C) (Figure 2). Wild southern lizards were observed on the surface at the narrowest range of surface temperatures (18–25 °C) (Figure 2). The wild northern-linage lizards were observed on the surface at the highest range of base-of-burrow relative humidity (70–97%) (Figure 3). Wild southern lizards were observed on the surface at the lowest range of base-of-burrow relative humidity (30–70%) (Figure 3).

The generalised additive model for wild lizards explained 43.8% of the variance in the data, with an adjusted R-square of 0.3 (Appendix A). We found moderate (*p* values between 0.05 and 0.01) to strong (*p* values between 0.01 and 0.001) evidence [61,62] for the relationships between approach distance in northern-latitude lizards and all three PC axes: PC1 (*p* = 0.02), PC2 (*p* = 0.01) and PC3 (*p* = 0.01) (Appendix A). Northern-lineage lizards decreased approach distance at mid-range values of PC1 (moderate temperature and humidity above and in the burrow) (Figure 4). The northern lizard reduction in approach distance in response to moderate temperature and humidity above and in the burrow was significantly different from the mid-range lineage, where approach distance increased under the same conditions (Appendix A). Northern-lineage lizards decreased approach distance as PC2 approached −40 (decreasing ambient temperature, increasing ambient humidity, decreasing mid-burrow humidity and increasing mid- and base-of-burrow temperature) (Appendix A). Northern-lineage lizards decreased approach distance as PC3 increased (high base-of-burrow humidity) (Appendix A). The northern lizard reduction in approach distance in response to high burrow humidity was significantly different from the mid-range lineage, where approach distance increased under the same conditions (Appendix A). Little or no evidence was found for relationships between the mid and southern lizards and the PC axes in the models for the wild sites (*p* > 0.1) (Appendix A).

### 3.5. Translocated Lizards

During the approach distance assays on translocated lizards, the northern lizards were observed on the surface at the broadest range of surface temperatures (20–45 °C) (Figure 2). Translocated southern lizards were observed on the surface at the narrowest range of surface temperatures (20–40 °C) (Figure 2). The northern lizards were observed on the surface at the narrowest range of burrow relative humidity (70–100%) (Figure 3 and Figure 5). Translocated southern lizards were observed on the surface at the broadest range of base-of-burrow relative humidity (27–100%) (Figure 3 and Figure 5). All three translocated lineages were observed on the surface under similar temperature and humidity conditions, in contrast to wild lineages (Figure 2 and Figure 3).

The generalised additive model for translocated lizards explained 29.1% of the variance in the data, with an adjusted R-square of 0.17 (Appendix A). We found moderate evidence for a relationship between approach distance in northern-lineage lizards and PC3 (*p* = 0.02) (Appendix A). Northern-lineage lizards showed a curvilinear response to PC3, with approach distances lowest in the mid-range of PC3 values (increasing base-of-burrow humidity) (Figure 5). The northern lizards’ increase in approach distance in response to high values of PC3 when burrow humidity was high was significantly different from the mid-range lineage, where approach distance decreased under the same conditions (Appendix A). Northern- and southern-lineage lizards differed significantly at low values of PC3, where burrow humidity was moderate to low: no northern lizards were observed under these conditions (Appendix A).

Little or no evidence was found for relationships between the mid and southern lizards and the PC axes in the models for the translocation (*p* > 0.1) (Appendix A).

## 4. Discussion

We monitored the behavioural responses to microclimatic variation in three latitudinally distinct pygmy bluetongue lineages in the wild and during their third activity season after translocation to a common southern location. Northern-lineage lizards differed from mid and southern lineages in how they responded to microclimatic variation, demonstrating significant links between behaviour and microclimate in the wild. Northern-lineage lizards reduced approach distance when surface conditions were moderate and burrow conditions were humid. This reduction was persistent in the common environment after two years, suggesting a plasticity limit or lag. Conversely, mid- and southern-lineage lizards showed a decoupling of behaviour and microclimate during unusually wet spring conditions. Results from the translocation site imply acclimatisation to the cooler conditions posed by climate change mitigation translocation may take more than two years and that acclimatisation time and/or ability may vary with latitude of origin.

### 4.1. Conditions for Reduced Approach Distance

Northern-lineage lizards in the wild adjusted approach distance with microclimate, remaining surface-active in the face of a potential predator under moderate (low temperature, high humidity) conditions. Consistent with the literature, northern lizards demonstrated a curvilinear response to temperature, with approach distances decreasing as temperatures reached an optimum [49,50]. In the first principal component axis, surface temperature and humidity were equally weighted and negatively correlated. As responses to temperature and humidity are inextricably linked [34,36,37,63,64,65], northern lizard behaviour was equally in response to a humidity optimum.

The northern lineage differed significantly from the mid-range lineage in the conditions that predicted reduced approach distance: moderate surface temperature and high surface and burrow humidity. As the mean and maximum temperature in the northern site were higher than in the mid-latitude site, the behavioural differences between these two lineages were not due to a lack of opportunity for northern lizards to be active at higher temperatures. The northern lizards’ reduced approach distance under low-temperature, high-humidity conditions is consistent with our previous findings for the southern-lineage lizards [18]. Consistently, lower burrow humidity at the southern site over both seasons combined with low-temperature, high-humidity surface activity supports hydroregulation as an optimal strategy for this lineage [36,66,67]. However, the northern site had the highest burrow humidity; therefore, hydroregulation does not explain the similarity between the southern and northern lineage approach behaviour.

Variable localised temperatures and selection for a low critical minimum may explain the narrow temperature and humidity range predictive of reduced approach distance in the northern lizards. The northern site had the most variable ambient temperature and humidity during this study. Temperature variation influences the thermal optima and the evolution of critical thermal minima, and may affect performance [4]. For example, multi-ocellated racerunners (*Eremias multiocellata*) showed altered behaviour and parturition date when exposed to variable temperatures versus a constant daily maximum [53]. Diel variability that regularly exceeds the thermal optimum may be accumulatively detrimental, and ectotherms may buffer environmental variation via thermoregulatory behaviours and by narrowing their performance range, as predicted by the diel narrowing hypothesis [5,68]. Optimality models predict selection for narrow performance ranges in environments with within-generation variability, as was seen for the northern site during our study [68]. Furthermore, in species like the pygmy bluetongue, small body size and low vagility may limit the extent of buffering via thermoregulation; therefore, a narrow performance range is favourable in variable conditions [68]. The climatic variability hypothesis predicts lower critical thermal minimum in more variable environments [69]. As the northern site was the most variable, northern lizards may have greater tolerance to lower temperatures than the other two lineages. It is therefore possible that the behavioural similarities between the northern and southern pygmy bluetongue lineages were due to different localised factors. These findings are consistent with previous work on this species suggesting latitudinally diverse lineages may be locally adapted to their microclimate [18]. Optimum performance under moderate conditions coupled with the ability to tolerate low temperatures would be advantageous when moving lizards southward to currently cooler conditions to mitigate climate change despite the change in latitude.

### 4.2. Conditions for Surface Activity

While northern lizards altered their behaviour in response to moderate conditions, they were observed on the surface at a broader range of surface temperatures and humidities than the other two lineages in the wild. The thermo-hydroregulation hypothesis [67] suggests thermoregulation accuracy should be increased if desiccation risk—and consequently the need for hydroregulation—is reduced due to increased moisture availability [37,67]. As our study was conducted during an unusually wet spring, it is likely that the northern lineage reduced hydroregulation and increased thermoregulation during the study period. Furthermore, the northern site had the highest mean burrow humidity of all the sites during this period despite having the lowest rainfall—indicating high-hydric-quality refuges—so the risk of desiccation was likely low [18,37]. If so, the northern lizards may have engaged in increased basking and active thermoregulation to reach and maintain optimum body temperatures. Reduced water availability leads to suppressed daily activity, basking, body growth and preferred body temperatures [33,34,36,63,70]. Therefore, the inverse—increased activity, higher preferred body temperature and increased investment in body growth due to the reduced desiccation risk during a wet spring—may have resulted in surface activity over a wide range of temperature and humidity conditions. Further monitoring of these three lineages under different seasonal conditions will provide insight into strategies for buffering seasonal variations in temperature and water availability, informative for climate change mitigation.

### 4.3. Relative Humidity in the Burrow

Wild northern-lineage lizards decreased approach distance, remaining surface-active in the face of a potential predator, with increasing burrow humidity. Burrows provide animals with thermal refuge, but they also provide refuge from hydric stress as relative humidity remains higher in the burrow than on the surface [18,36,70,71]. The link between approach distance and burrow humidity is consistent with our previous work, which found burrow humidity to be a strong predictor of pygmy bluetongue approach distance behaviour [18]. While mean burrow humidity was highest in the northern site, the northern-lineage lizards showed the same approach distance response and were only observed surface-active when burrow humidity was high in both the wild and the translocation site. In contrast, the translocated southern and mid lineages were observed on the surface across a broader range of burrow humidity. Lineage differences in hydroregulation can persist in altered humidity conditions [30,34], and acclimations to laboratory-altered temperatures may take greater than four months [66]. It is therefore likely that lineage differences in hydroregulation may persist in the wild beyond one or two activity seasons. The persistent link between burrow humidity and behaviour seen in the northern-lineage lizards supports our earlier findings that pygmy bluetongues may have local adaptations to their site of origin [42] and suggests that northern lizards have limited plasticity after two seasons of acclimation to the common environment. Monitoring for another season will help clarify if the northern lineage will fully acclimatise to the translocation by demonstrating behavioural responses to microclimate in keeping with the other two lineages.

### 4.4. Acclimatisation to Translocation

Ectotherms demonstrate intraspecific variation in plasticity limits, often latitude- or altitude-based; therefore, understanding plasticity on the population/lineage level is imperative for climate change mitigation [12,18,72,73]. For species such as the pygmy bluetongue where translocation polewards is the most favourable climate change mitigation, understanding plasticity extents, limits, and timeframes for acclimatisation are critical to species survival [45,48,72]. Tolerances to high temperatures in populations of squamates are thought to be locally adapted or “fixed” [10,74,75,76,77]. Conversely, tolerance to cooler temperatures such as those presented by translocation for climate change mitigation are thought to have more plasticity [10,74,76,78]. As thermoregulation is not as effective at buffering cold as it is heat, some degree of acclimatisation to the translocation environment is required [17,79]. Contrary to our previous findings from the first season of the translocation, the mid and southern lineages in the third season of translocation did not differ significantly from each other and neither lineage showed significant adjustment to approach distance in response to the microclimate variables [18]. During the behavioural assays, the mid and southern lineages were observed on the surface under very similar conditions in the translocation. Acclimation studies show adjustments to cooler temperatures may take upwards of four months in a laboratory; therefore, they may require more than one activity season in a field setting [17,18,66]. It is therefore likely that the similarity among the mid and southern lineages in the third year of translocation represents a gradual acclimatisation to these cooler conditions.

Squamate species show spatial population variance in critical thermal minima [10,80,81]. If the latitudinally diverse translocated lineages have different critical thermal minima when they enter the translocation, acclimatisation time in a common environment may vary from lineage to lineage. For example, while the higher critical thermal minimum of the warm-climate population persisted in green anoles (*Anolis carolinensis*), relative rates of acclimation were comparable between the cool-climate and warm-climate population individuals, with critical thermal minimum decreasing by 4.5 ± 2.4 °C in both groups over two weeks [17]. The persistent response of the northern lineage to burrow humidity seen in the translocation implies that a longer timeframe is required for this lineage to adjust to the cooler environment or that the trait is genetically fixed, and further research is warranted on the genetic basis of this trait.

In the current study, the wild mid and southern lineages did not differ in their response to microclimatic variation. Therefore, similarities in behaviour between these lineages at the translocation site was not unexpected. However, despite the common conditions at the translocation site, some lineage-specific behaviours persisted; the southern lineage was observed surface-active at low to moderate burrow humidities both in the wild and in the translocation. The retention of site-of-origin thermal behaviours may result in alterations to hydro-thermal regimes [47,82]. Prolonged basking to reach unattainable preferred temperatures increases predation risk, reduces daily activity time and may have knock-on effects such as prolonged gestation, reduced recruitment and altered litter size [83,84,85,86,87,88]. If the lineages continue to respond more similarly to their site of origin than to each other in the translocation, further work is urgently needed to determine potential detrimental effects that may limit long-term establishment at new sites. Retention of site-of-origin behaviour may be eliminated [82,89] or persist [90,91] in offspring gestated in the new environment, and maternal effects may influence behaviour of offspring for more than one generation [92,93,94]. Future work should investigate the hydro-thermal behaviours of offspring born in the translocation site to determine if transgenerational plasticity and/or maternal effects may affect establishment of this species in mitigation translocation environments.

### 4.5. Lineage Response to Microclimate

The mid and southern lineages did not adjust approach behaviour in response to microclimate in this study. This finding is contrary to our previous work where the mid and southern pygmy bluetongue lineages showed behavioural responses to localised microclimate in the wild that persisted in the translocation [18]. Numerous variables other than microclimate may influence lizard behaviour and thus it is often difficult to pin down possible predictive factors [37,47,70,95]. The unusually heavy rainfall experienced during the current study may have obscured any relationships between behaviour and microclimate in the mid and southern lineages. Increased vegetation and higher prey availability may also affect behaviour after rainfall; however, these factors have time lags and are therefore unlikely to have shown measurable effects during the three months of this study [96]. In contrast, increased water availability and humidity directly affect behaviour in lizards [29,33,34,37,64,65,70,97]. Furthermore, stochastic weather events have more pronounced effects on the behaviour of low-dispersal organisms such as pygmy bluetongues [97,98]. Our previous work on this species showed evidence of hydroregulation in a dry year [18], but in this study, behaviour largely decoupled from the microclimate in two of the three lineages. This decoupling of behaviour and microclimate is in accordance with the thermo-hydroregulation hypothesis, which predicts a decrease in hydroregulation and an increase in active thermoregulation when desiccation risk is reduced [37,67]. This result is consistent with other studies that have demonstrated increased activity in the eastern fence lizard (*Sceloporus undulatus*) and the bronze anole (*Anolis aeneus*), as well as higher growth rates in the common lizard (*Lacerta vivipara*) and the tree dtella (*Gehyra variegata*) in response to high rainfall [31,33,34,63]. Further monitoring of the lineages in the wild and in the translocation during another drier season would provide clarity on behavioural differences across the three lineages, behavioural alterations in wet versus dry seasons and long-term behavioural acclimatisation periods for translocations when mitigating climate change effects.

## 5. Conclusions

Our results show that latitudinally varied lineages of pygmy bluetongue lizards differed in their behavioural response to microclimatic variation. Northern-lineage lizards showed significant links between behaviour and microclimate in the wild, remaining surface-active in the face of a predator when conditions were moderate. Variable microclimate conditions in the northern site may have resulted in a narrow optimal performance breadth, tolerance to cooler temperatures and a lower critical minimum for the northern lineage. If so, the ability to perform optimally in cooler conditions would be favourable for establishment in southern translocation sites to mitigate predicted climate change effects. These findings highlight the importance of including climatic variability when investigating response to microclimate and have positive implications for mitigation planning for the northern lineage.

Some evidence for acclimatisation was seen after two full activity seasons in the translocation environment, implying pygmy bluetongue lineages take more than one activity season to adjust to the changed conditions posed by climate change mitigation translocation. Conversely, some behavioural responses to microclimatic variation were retained in the northern lineage, suggesting a plasticity lag or limit for this lineage, as well as variation in acclimatisation time and/or ability with localised critical minima and/or latitude of origin. Another monitoring season would help clarify if the lineages are acclimatising. As this study focused on adult lizards, an investigation into behavioural responses of offspring produced in the translocation will increase our understanding of plasticity extents and limits.

High spring rainfall likely led to a decoupling of behaviour and microclimate in the mid and southern lineages. The results support the thermo-hydroregulation hypothesis, with a reduction in hydroregulation during wet spring conditions [37,67]. The difference between behaviour in this wet spring compared to the previous drier season [18] highlights the importance of long-term monitoring over ecologically appropriate timescales and covering a variety of seasonal variations when assessing translocation acclimatisation in wild animals. Further monitoring of the lineages in the wild and in the translocation during another drier season would provide clarity on behavioural differences across the three lineages, behavioural alterations in wet versus dry seasons and long-term behavioural acclimatisation periods for translocations when mitigating climate change effects.

Australia has the highest reptile diversity in the world, coupled with high biodiversity loss [99,100]. Translocation is becoming increasingly urgent for the conservation of numerous reptile species [101]. When mitigating climate change via translocation, animals require movement to future-suitable sites. Therefore, minimising latitudinal and microclimate differences between source site and translocation site is not practical. While our results support our previous study, which found that latitudinally diverse pygmy bluetongues lineages may vary in their response to microclimate, we suggest that variation may not always be detrimental and may potentially aid animals in colonisation in changed environments. Results from this study are directly applicable to the ongoing conservation management of this species and applicable for the management of small burrow-dwelling reptiles in general.

## Figures and Tables

**Figure 1 biology-14-00622-f001:**
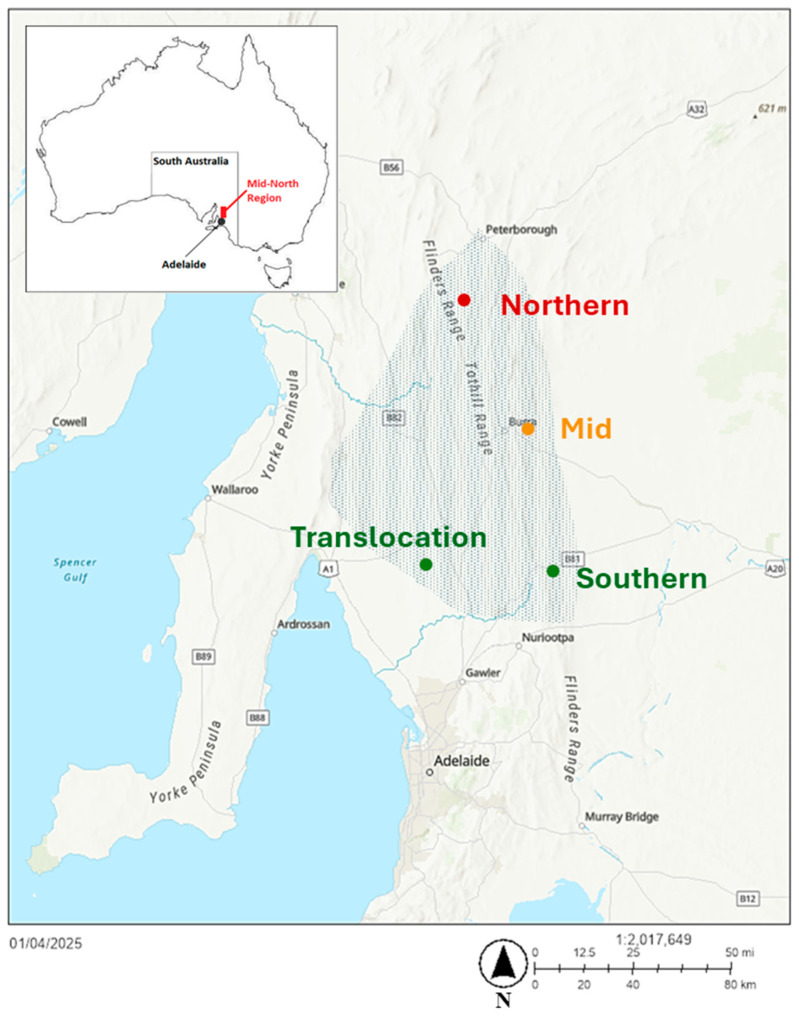
Map showing locations of pygmy bluetongue (*Tiliqua adelaidensis*) source sites—northern (Jamestown), mid (Burra) and southern (Kapunda)—and the translocation site near Tarlee, with a red rectangle representing the Mid North region on the insert map. The shaded section represents the current species range.

**Figure 2 biology-14-00622-f002:**
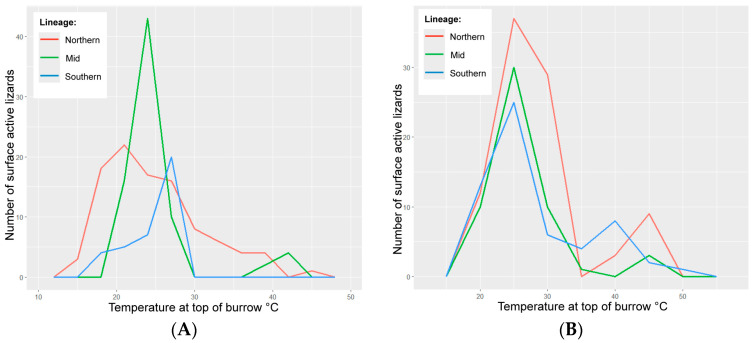
Surface activity curves for northern, mid and southern lineages of pygmy bluetongue lizard *(Tiliqua adelaidensis*) showing the total number of lizards that were surface-active—observed on the surface during approach distance assays—at a given temperature (°C) in 6 °C “bins” or increments, where panel (**A**) shows the results from the wild sites and panel (**B**) shows the results from the translocation site.

**Figure 3 biology-14-00622-f003:**
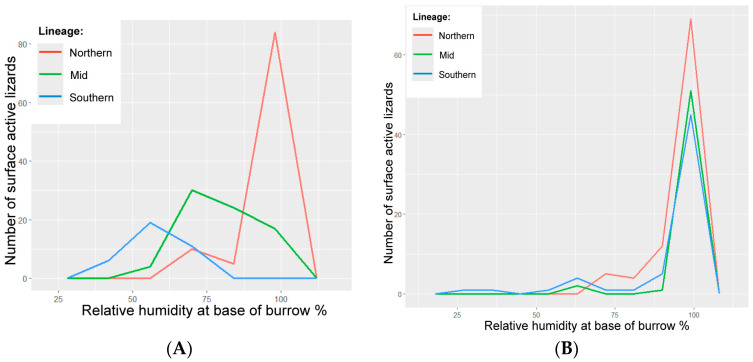
Surface activity curves for northern, mid and southern lineages of pygmy bluetongue lizard (*Tiliqua adelaidensis*) showing the total number of lizards that were surface-active—observed on the surface during approach distance assays—when the base of the burrow was at a given relative humidity (%) in 8% “bins” or increments, where panel (**A**) shows the results from the wild sites and panel (**B**) shows the results from the translocation site.

**Figure 4 biology-14-00622-f004:**
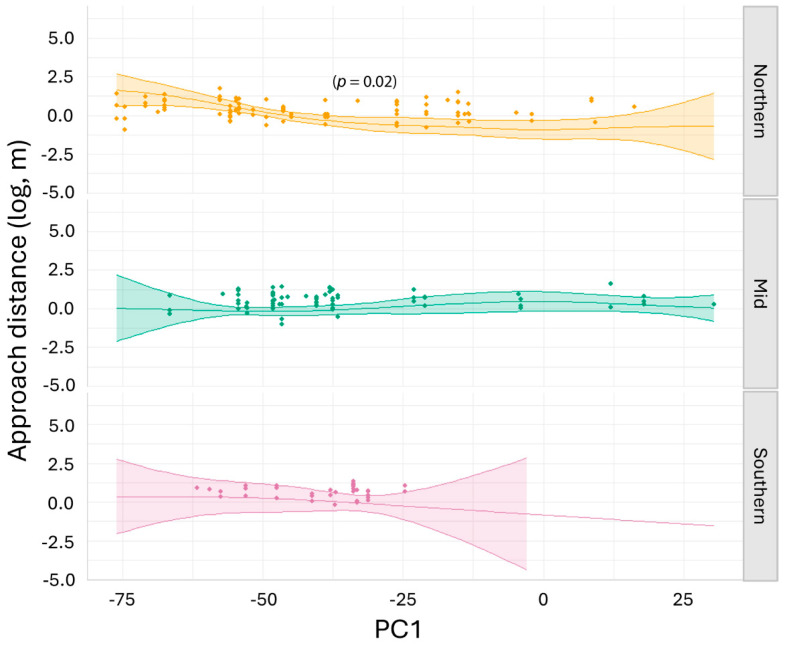
Model of predicted approach distance (log, m) and raw data points for northern (Jamestown), mid (Burra) and southern (Kapunda) lineages of pygmy bluetongue lizard (*Tiliqua adelaidensis*) in the wild for PC1 (increasing temperature and decreasing relative humidity as PC1 values increase) and lineage. Lines depict model predictions, coloured bands depict 95% C.I and points depict the raw data.

**Figure 5 biology-14-00622-f005:**
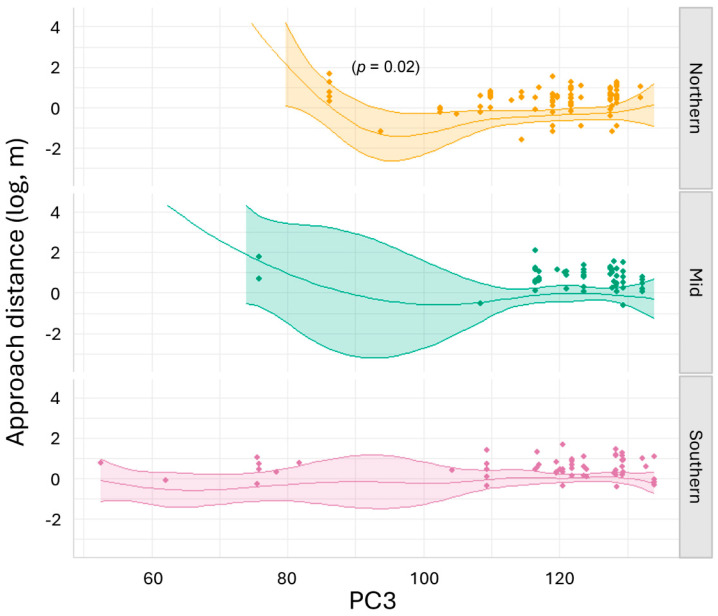
Model of predicted approach distance (log, m) and raw data points for northern (Jamestown), mid (Burra) and southern (Kapunda) lineages of pygmy bluetongue lizard (*Tiliqua adelaidensis)* in the translocation for PC3 (increasing burrow relative humidity as PC3 values increase) and lineage. Lines depict model predictions, coloured bands depict 95% C.I and points depict the raw data.

**Table 1 biology-14-00622-t001:** PCA rotations and respective component loadings for environmental variables, temperature °C and relative humidity % above the burrow (ambient) and in the burrow (top, middle and base) from October 2022 to April 2023 at all sites (northern (Jamestown), middle (Burra), southern (Kapunda) and the translocation near Tarlee). Bold indicates contributions > 0.30 in the first 3 principal component axes to aid biological interpretation of these 3 PC axes.

	PC1	PC2	PC3
Mean ambient temperature	**0.36**	**−0.50**	−0.09
Mean ambient relative humidity	**−0.36**	**0.46**	0.13
Mean top-of-burrow temperature	**0.42**	−0.18	0.00
Mean top-of-burrow relative humidity	**−0.40**	−0.01	0.02
Mean mid-burrow temperature	**0.40**	**0.30**	0.15
Mean mid-burrow relative humidity	**−0.32**	**−0.44**	0.16
Mean base-of-burrow temperature	**0.36**	**0.44**	0.22
Mean base-of-burrow humidity	0.00	−0.19	**0.94**

## Data Availability

The original data presented in this study are openly available on ROADS at 10.25451/flinders.28537187.

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
