# Peer review of "Lizards, Lineage and Latitude: Behavioural Responses to Microclimate Vary Latitudinally and Show Limited Acclimatisation to a Common Environment After Two Years"

_biology, 2025, doi:10.3390/biology14060622_

Round 1
Reviewer 1 Report
Comments and Suggestions for Authors
Trewartha and colleagues examined the effects of population and refuge microclimate on avoidance behavior in a lizard. The study system is neat and well-suited for some of the authors’ goals. I thought the behavior assay was clever, and I appreciated the effort to connect the work with thermal biology theory. I think including information about the optimal temperature and/or preferred temperature for individuals in each location would strengthen this connection.
However, I was not convinced that latitude affected results very much—the study locations were very close to one another and did not vary much in the microclimates of the burrows. I think further explanation is needed in the materials and methods, especially regarding the measurement of the burrows, sample sizes, and characteristics of the study animals. Last, the Introduction and Discussion are too long due to verbose writing and redundancy. I think the manuscript would be much-improved if these sections were shorter and more tightly written.
Comments:
Simple summary, general comment: Given the admitted bias of work in the northern hemisphere, it may be worth referring to northerly sites as low latitude sites (and southerly sites as high latitude). It makes the ideas more generalizable.
L29-30: This is debatable, but Class Reptilia is far from the most biodiverse class of ectotherms. Class Insecta alone contains <50% of ectothermic species. Even among vertebrates, Class Reptilia is not the most diverse—most species of ectothermic vertebrates are fishes. I’d remove this sentence and re-word the previous sentence as “Ectotherms, including reptiles, are at particular risk…”
L61-63: This sentence can be removed, and the 2nd and 3rd paragraphs can then be combined.
Introduction, general comment: It is very long. I recommend condensing the ideas of the last three paragraphs.
L160: Were all of the lizards monitored in enclosures? Or just the translocated lizards? Trials were run 2 years after translocation, so did all of the translocated lizards survive? Given this time line, is it possible that the translocated animals responded differently because they were older (on average) than the wild lizards? How might age, body size, sex, etc. affect results?
Figs. 2-7: Very difficult to read due to gray and small font.
L199-201: Just to be clear, it seems like the artificial burrows were used to estimate the environmental conditions of the burrows actually used by the lizards in real-time. Is this correct? If not, please explain this approach in more detail. If this was the approach, was there a validation done showing that artificial and natural burrows at each site exhibited similar values?
L233-243: There is quite a bit of discussion about latitude, but results here do not necessarily reflect the expected differences (higher temperature and, thus, lower RH at northern site). Given these results, maybe just refer to them as representative sites, rather than N, mid, and S.
L245: How many burrows at each site were monitored? How many lizards at each site were monitored?
Figs. 2 and 3: These two figures best illustrate interesting story points as differences among the lineages largely disappear in a common garden.
Fig. 3: How can RH exceed 100%?
Figs. 6 and 7: Are these figures necessary? PC3 seems to be nearly fully redundant with RH trends shown in Fig. 3.
Figs. 4-7: It might be nice to have some indication on the figure panels for which relationships are significant (e.g., p values in the top right corner for each location).
L346-347: Please confirm that other aspects did not drive these differences. For example, age, sex, or size of study individuals may have varied across the locations, and these factors may have influenced results more than location.
L369, 399-408: Do the authors have information regarding these lizards’ preferred temperatures or optimal body temperatures? This would important when making predictions about the optimality of burrow microclimates. The idea of Topt is introduced here, but it is not clear it’s known (or if it varies among the populations studied). Without this information, it is difficult to understand the ecological relevance of shifts in approach distance relative as they relate to burrow microclimate.
Author Response
Thank you for your comments. Please see attached word file for our responses.

Reviewer 2 Report
Comments and Suggestions for Authors
Dear Authors,
the manuscript investigates very interesting and important issue. The introduction provides a good starting point to your research. However, there are certain issues that need to be corrected prior to the publication of your manuscript.
In the simple summary and abstract it is unclear how many sites you investigated, and how many populations. As well, it is unclear how this translocation site looks like- whether it is a wild site or an enclosure in captivity? How do you know in this translocation site which animals are from which lineage? Generally, translocation would mean that the animals are intorduced to another site in the wild, and here it seems that this is not full wilderness, but something in between?! Please, clarify all this.
Line 144 you say that 4 sites were investiated and in the short summary and abstract you mention three sites... so this whole thing is very confusing. Please, explain clearly how many sites and expalin this translocation site more in detail.
In M&M lines 209-214 are the results of your PCA and not methods, as well as table 1.
Discussion section is too lenghthy, and many things are redundant from the introduction and the results. Please, shorten the whole discussion and stick to the discussion, and not providing all the results in details, since they are already provided in the results section.
Author Response

(The authors gave the same response as above.)

Reviewer 3 Report
Comments and Suggestions for Authors
This manuscript addresses interesting topics; however, I suggest the authors consider the following observations and comments:
The research question or objectives are unclear; I believe the authors should close section 1 with specific and concrete questions. This will greatly help avoid confusion for the reader.
Paragraphs 177-191. The authors use a lot of rhetoric; I believe they should precisely indicate how they measured the approach distance. Paragraphs 177-181 might be better suited to the introductory section.
Figure 1. Indicate geographic north. I suggest the authors create the map with ArcGIS or QGIS to provide a better view for readers.
Figures 2-7. Make the font larger; the reader will have great difficulty understanding the text.
I suggest authors include Barry Sinervo and Ray Huey in their references (the latter is in reference no. 69, line 797). They have several publications that can strengthen the introduction and discussion, particularly between lines 341 and 343. For example:
Sinervo, B., Mendez-De-La-Cruz, F., Miles, D. B., Heulin, B., Bastiaans, E., Villagrán-Santa Cruz, M., ... & Sites Jr, J. W. (2010). Erosion of lizard diversity by climate change and altered thermal niches. Science, 328(5980), 894–899.
Sinervo, B., Miles, D. B., Wu, Y., MÉNDEZ‐DE LA CRUZ, F. R., Kirchhof, S., & Qi, Y. (2018). Climate change, thermal niches, extinction risk and maternal‐effect rescue of toad‐headed lizards, Phrynocephalus, in thermal extremes of the Arabian Peninsula to the Qinghai—Tibetan Plateau.Integrative zoology, 13(4), 450-470.
Sinervo, B., Reséndiz, R. A. L., Miles, D. B., Lovich, J. E., Rosen, P. C., Gadsden, H., ... & de la Cruz, F. R. M. (2024). Climate change and collapsing thermal niches of desert reptiles and amphibians: Assisted migration and acclimation rescue from extirpation.Science of the Total Environment, 908, 168431.
Huey, R. B., Deutsch, C. A., Tewksbury, J. J., Vitt, L. J., Hertz, P. E., Álvarez Pérez, H. J., & Garland Jr, T. (2009). Why tropical forest lizards are vulnerable to climate warming.Proceedings of the Royal Society B: Biological Sciences, 276(1664), 1939-1948.
Huey, R. B., Kearney, M. R., Krockenberger, A., Holtum, J. A., Jess, M., & Williams, S. E. (2012). Predicting organismal vulnerability to climate warming: roles of behavior, physiology and adaptation.Philosophical Transactions of the Royal Society B: Biological Sciences, 367(1596), 1665-1679.
Wild, K. H., Huey, R. B., Pianka, E. R., Clusella-Trullas, S., Gilbert, A. L., Miles, D. B., & Kearney, M. R. (2025). Climate change and the cost-of-living squeeze in desert lizards.Science, 387(6731), 303-309.
Author Response

(The authors gave the same response as above.)

Reviewer 4 Report
Comments and Suggestions for Authors
This manuscript examines the geographic variation in approach distance of pygmy bluetongue lizards near their burrows. It also compares wild with translocated individuals. The study also explores the potential environmental correlates of approach distance. The topic is interesting and the study generally well done. I have a few suggestions to improve the manuscript.
Lines 33/34 and 37/38: Why the extra space between lines in the Abstract?
Lines 100 & 101: Should be “speciose” not “specious”
Line 149: add “site” after “translocation”
Lines 195-197: How similar are the artificial burrows to natural burrows? In other words, is there any evidence that the artificial burrows are representative of natural burrows?
Lines 231-232: I would put this information in the Table caption.
Lines 253-256: Any analyses to support this conclusion or is it just a qualitative assessment?
Lines 275, 320: Why not use r2 instead of “R-squared”?
Lines 275-276: How do you define moderate to strong evidence?
Lines 312-315: How do you know this? Did you perform a rigorous analysis?
Line 318: Should it be “Figure 3, 7”?
Lines 349-351: Move this to right after the sentence reporting the Northern latitude result (lines 345-347).
Line 356: What was the significant adjustment?
Lines 420-421: However, you have not really provided any specific evidence for this different beyond a qualitative comparison.
Line 444 and elsewhere: Could decreasing approach distances reflect a more cryptic strategy rather than “prioritising surface activity”?
Author Response

(The authors gave the same response as above.)

Round 2
Reviewer 1 Report
Comments and Suggestions for Authors
I appreciate the authors for incorporating feedback. The revised manuscript is much-improved.
Author Response
(reviewer comment) I appreciate the authors for incorporating feedback. The revised manuscript is much-improved.
(author response) Thank you very much for your time and assistance in improving our manuscript. It is much appreciated.
Reviewer 2 Report
Comments and Suggestions for Authors
Dear authors,
Data provided on translocation and other methods used in the study give enough information to clearly understand the experiments conducted in this research. I have some minor suggestions to still improve the manuscript so that it would be easier to follow.
In the results section, I would start with environmental analyses and leave the PCA results for later as this sounds as more logical flow.
Line 263- delete repeated words (at a at a given…)
Regarding discussion and conclusion, although these parts are shortened for some 800 words, they are still too long. And again, too many details on the results are presented, that should not be repeated in such detail.
Author Response
Comment 1: In the results section, I would start with environmental analyses and leave the PCA results for later as this sounds as more logical flow.
Response 1: Thank you this is a great suggestion, we have put the environmental variables section at the start of the results and the PCA results before the analysis results.
Comment 2: Line 263- delete repeated words (at a at a given…)
Response 2: Thank you for pointing this out, it has been fixed.
Comment 3: Regarding discussion and conclusion, although these parts are shortened for some 800 words, they are still too long. And again, too many details on the results are presented, that should not be repeated in such detail.
Response 3: Thankyou for your help with improving our manuscript. We have removed several repetitive result summary sentences throughout and reduced the overall discussion by a further 160 words.
Reviewer 3 Report
Comments and Suggestions for Authors
The authors made a great effort to address the comments and suggestions made to first version of the manuscript. This second version makes the document more robust and well-founded.
Author Response
(reviewer comment) The authors made a great effort to address the comments and suggestions made to first version of the manuscript. This second version makes the document more robust and well-founded.
(author response) Thank you very much for your time and assistance in improving our manuscript. It is much appreciated.